# The Value of Flow Cytometry Clonality in Large Granular Lymphocyte Leukemia

**DOI:** 10.3390/cancers13184513

**Published:** 2021-09-08

**Authors:** Valentina Giudice, Matteo D’Addona, Nunzia Montuori, Carmine Selleri

**Affiliations:** 1Department of Medicine, Surgery, and Dentistry, Scuola Medica Salernitana, University of Salerno, 84081 Baronissi, Italy; vgiudice@unisa.it (V.G.); matteo.daddona.fuqm@na.omceo.it (M.D.); 2Hematology and Transplant Center, University Hospital “San Giovanni di Dio e Ruggi d’Aragona”, 84131 Salerno, Italy; 3Department of Translational Medical Sciences, “Federico II” University, 80131 Naples, Italy; nmontuor@unina.it

**Keywords:** T large granular lymphocytic leukemia, clonality, flow cytometry, Vβ usage

## Abstract

**Simple Summary:**

Large granular lymphocyte (LGL) leukemia, a lymphoproliferative disease, is characterized by an increased frequency of large-sized lymphocytes with typical expression of T-cell receptor (TCR) αβ, CD3, CD8, CD16, CD45RA, and CD57, and with the expansion of one to three subfamilies of the TCR variable β chain reflecting gene rearrangements. Molecular analysis remains the gold standard for confirmation of TCR clonality; however, flow cytometry is time and labor saving, and can be associated with simultaneous investigation of other surface markers. Moreover, Vβ usage by flow cytometry can be employed for monitoring clonal kinetics during treatment and follow-up of LGL leukemia patients.

**Abstract:**

Large granular lymphocyte (LGL) leukemia is a lymphoproliferative disorder of mature T or NK cells frequently associated with autoimmune disorders and other hematological conditions, such as myelodysplastic syndromes. Immunophenotype of LGL cells is similar to that of effector memory CD8^+^ T cells with T-cell receptor (TCR) clonality defined by molecular and/or flow cytometric analysis. Vβ usage by flow cytometry can identify clonal TCR rearrangements at the protein level, and is fast, sensitive, and almost always available in every Hematology Center. Moreover, Vβ usage can be associated with immunophenotypic characterization of LGL clone in a multiparametric staining, and clonal kinetics can be easily monitored during treatment and follow-up. Finally, Vβ usage by flow cytometry might identify LGL clones silently underlying other hematological conditions, and routine characterization of Vβ skewing might identify recurrent TCR rearrangements that might trigger aberrant immune responses during hematological or autoimmune conditions.

## 1. Introduction

Large granular lymphocyte (LGL) leukemia is a chronic lymphoproliferative disorder arising from clonal expansion of mature T or Natural Killer (NK) cells, and accounts for 2–5% of all cases of non-Hodgkin lymphomas (NHL) in Western countries [1]. LGL leukemia is considered a bone marrow failure (BMF) syndrome because of its overlapping pathogenesis with other immune-mediated diseases [2], and because LGLs can be frequently found in patients diagnosed with BMF syndromes, such as pure red cell aplasia (PRCA) and acquired aplastic anemia (AA) [3]. LGL leukemia of mature T cells is the most common entity (85% of cases), usually occurring with an indolent clinical course, while NK-LGL leukemia is rare but aggressive and frequently associated with Epstein–Barr virus (EBV) infection in young Asian adults [1,2,3,4,5,6,7,8]. LGLs are large-sized (15–18 µm) lymphocytes with an abundant cytoplasm containing azurophilic granules, and a round or reniform nucleus with mature chromatin [1]. LGLs can also be found in reactive conditions; however, circulating granular lymphocyte frequency is typically < 300 cells/µL, while LGL count is > 2000 cells/µL during LGL leukemia [1,4,5,9]. In the majority of cases, leukemic cells display a characteristic phenotype with surface expression of TCR αβ, CD3, CD5^dim^, CD8, CD16, CD45RA, and CD57, while cells are negative for CD4, CD27, CD28, and CD45RO, resembling effector memory and terminal effector memory T cell phenotype [1,7,10]. In other variants, neoplastic cells can be CD3^+^CD4^+^CD8^dim^CD57^+^ or CD3^+^CD4^+^CD8^+^CD57^+^ with various expression of CD56, CD57 and CD16, and in fewer cases leukemic cells are TCR γδ^+^ [11,12,13,14,15]. In NK-LGL leukemia, LGLs are positive for CD2, CD3ε, CD8, CD16, and CD56, while negative for sCD3, CD4, and TCR αβ [16]. T- and NK-LGL cells show restriction (oligoclonality) of T-cell receptor (TCR) or killer immunoglobulin-like receptor (KIR) repertoire that can be identified by flow cytometry, polymerase chain reaction (PCR), or next-generation sequencing (NGS) [17]. Clinical features are not specific to LGL leukemia, as the most common manifestations are neutropenia, anemia, and splenomegaly usually occurring asymptomatically [1,18,19,20]. In case of severe neutropenia (absolute neutrophil count < 0.5 cells/L), patients can experience recurrent oral ulcerations and infections, especially bacterial, less frequently viral and fungal infections [6,19,21]. In 6–22% of cases, anemia can be severe (hemoglobin < 8 g/dL) and require transfusions, and can be associated with autoimmune hemolytic anemia (AIHA) or PRCA [6]. In addition, LGL leukemia patients frequently have a history of autoimmune disorders, such as rheumatoid arthritis (RA), or autoimmune endocrinopathies (e.g., thyroiditis) [19,22,23]. Conversely, CD4^+^ or CD4^+^CD8^+/−^ T-LGL leukemia has an indolent course and is frequently characterized by lymphocytosis with normal hemoglobin and platelet count and by the presence of a second neoplasms, especially B-cell NHL [24,25].

In this review, we focus on biological significance of LGL clonality and clinical utility of TCR repertoire investigation by flow cytometry.

## 2. T-LGL Leukemia Pathogenesis

The exact pathogenesis of LGL leukemia is still unclear. The most accepted hypothesis is a clonal drift of a T cell population after chronic antigen exposure, as suggested by cross-reactivity to human T-cell lymphotropic virus 1 (HTLV-1) epitopes or a frequent concomitant chronic viral infection in LGL leukemia patients, especially EBV, hepatitis C virus (HCV), or cytomegalovirus (CMV), especially in CD4^+^ T-LGL [1,26,27,28,29]. Despite there being no conclusive evidence that LGLs are activated by known viral antigens, immunohistochemical studies report clusters of LGLs in close contact to bone marrow (BM)-resident dendritic cells (DCs), supporting the hypothesis of a chronic self-antigen stimulation [18,30,31]. Once activated, LGLs expand under interleukin (IL)-15 and platelet-derived growth factor (PDGF) stimulation. IL-15, a proinflammatory cytokine, is overexpressed in LGL leukemia and drives clonal transformation of normal activated cytotoxic T cells, likely through centrosome alterations, aneuploidy, and increased resistance to apoptosis [2,32,33,34]. In addition, IL-15 induces transcription of several anti-apoptotic proteins, such as Bcl-2, while increases proteasome-mediated degradation of pro-apoptotic factors, such as Bid, resulting in increased cell survival [35]. IL-15 also upregulates *MYC*, *AURKA*, and *AURKB*, responsible for centrosome alterations, and *DNMT3B*, ultimately leading to hypermethylation of tumor suppressor genes [36,37]. However, pathogenesis of LGL leukemia is not driven only by chronic antigen and proinflammatory cytokine stimulation, but also by constitutive activation of Janus kinase (JAK)-signal transducer and activator of transcription (STAT) signaling pathway and resistance to Fas/Fas-ligand (Fas-L)-mediated apoptosis [1,2,38]. IL-6, another proinflammatory cytokine increased in the sera of LGL leukemia patients, is the main activator of STAT3, and is supposed to be mostly released by DCs [39,40]. Upon chronic IL-6 stimulation, LGLs show augmented expression of Mcl-1, an anti-apoptotic protein of the Bcl-2 family [40]. On the other hand, somatic mutations in *STAT3* gene can also induce constitutive activation of JAK/STAT pathway, as well as mutations on the *STAT5b* gene, more frequently described in CD4^+^ T-LGL leukemia [41,42,43]. Soluble Fas-L (sFas-L), increased in the sera of LGL leukemia patients, especially those with CD8^+^CD16^+^CD56^−^ LGLs and STAT3 hyperactivation, functions as a decoy receptor preventing Fas-mediated apoptosis [38,44,45,46]. FasL is also involved in development of neutropenia and its release is STAT3-dependent. Indeed, FasL expression is highly related to STAT3 activation status and CD8 LGL phenotype, while negatively influenced by miR-146b expression, frequently elevated in the plasma of AA and negatively correlated with absolute reticulocyte count [47,48]. In LGL leukemia, the pro-/anti-apoptotic balance is impaired in favor of increased resistance to cell death by neoplastic clones, also mediated by enhanced phosphoinositide 3-kinase (PI3K)/Akt signaling pathway activation through RANTES (also known as chemokine (C-C motif) ligand 5 or CCL5), IL-18, and macrophage inflammatory protein (MIP)-1b stimulation [33,39]. In addition, activation of tumor necrosis factor–related apoptosis-inducing ligand (TRAIL) receptor induces NF-κB signaling pathway, contributing to resistance to apoptosis in leukemic cells [49]. Neutropenia is a frequent clinical finding in LGL leukemia patients; however, pathogenetic mechanisms are still unclear and likely involve peripheral or intramedullary neutrophil destruction, such as FasL-mediated neutrophil apoptosis [5,50]. Indeed, the BM is frequently hypercellular with a left-shifted myeloid maturation, while less often hypocellular with reduced mature neutrophils, suggesting a compensatory myelopoiesis because of increased destruction [50].

## 3. TCR Clonality

Reactive LGLs can be found at low frequencies in healthy subjects after viral infections; however, in these cases, LGLs are polyclonal while neoplastic clones are oligo- or monoclonal [1]. The terms “polyclonal” or “oligo/monoclonal” refer to the diversity of TCR or B-cell receptor (BCR) repertoire in T- or B-cell pool of an individual, respectively [51,52]. For T lymphocytes, oligoclonality is defined based on the diversity of TCR repertoire based on distribution of variable (V) β and/or α chain rearrangements, or on skewing of a particular region namely the complementarity-determining region (CDR) 3 [53,54,55,56]. In other words, clonality refers to the ability of a random pool of T cells, either CD4^+^ or CD8^+^ lymphocytes, to recognize a large number of antigens (polyclonality) or limited ones (oligoclonality) [57]. TCR Vβ repertoire can be infinitely diverse because T cells can theoretically recognize any known or novel self- and non-self-antigens. The majority of T cells carry a TCR composed by an α and a β chain assembled after a complex gene locus rearrangement known as V(D)J rearrangement [57,58,59,60,61,62,63,64]. In this process, a random Vβ locus out of 52 available (grouped in 22 functional families) is sequentially rearranged with the Dβ1 locus, a random Jβ1 out of six, a Cβ1 and a Dβ2 loci, a random Jβ2 region out of seven available, and a Cβ2 locus. After this complex rearrangement, additional random modifications are added by a terminal deoxynucleotidyl transferase (TdT) enzyme between the end of the Vβ1 and the beginning of Jβ1 (the CDR3 region) [62,63,64]. V(D)J recombination and TdT modifications ideally produce up to 10^18^ diverse TCRs meaning that an individual might recognize up to 10^18^ diverse antigens [65]. In early phases of infections, polyclonal CD4^+^CD28^+^ and CD8^+^CD28^+^ T cells are activated and expand to increase the chance of having a functional antigen-specific immune clone that can fight pathogens [66,67]. If this clone can efficiently clear the infectious agent, high antigen-specific CD8^+^CD28^−^CD57^+^ effector memory T cells become immunodominant and constitute an oligoclonal pool of memory T cells [67]. Oligoclonal expansion of CD8^+^ T lymphocytes is observed in several autoimmune diseases and cancers and is related to poor survival [68,69]. Moreover, oligoclonality strongly suggests an antigen-driven T cell activation, as also described in AA, where CD8^+^ effector memory cells show oligoclonality with 1–3 immunodominant clones private to the disease [53,70,71,72]. Those clonotypes can be also present at very low frequencies in healthy individuals, supporting the hypothesis of an immune response to common epitopes [53]. Interestingly, LGL phenotype resembles the immunophenotype of effector memory T cells with CD3, CD4 or CD8, CD57 positivity, clonal expansion of a particular immunodominant clone, and gene expression profiling similar to that reported in healthy counterpart [73]. TCR clonality mirrors the oligoclonal drift that characterized T-LGL cell expansion likely driven by chronic antigen stimulation, IL-15 and PDGF overstimulation, and acquisition of somatic mutations in *STAT3* or *STAT5b* genes [1,32,33]. Despite common epitopes are believed to trigger autoimmune responses against hematopoietic stem and progenitor cells (HSPCs) in T-LGL leukemia and other marrow failure syndromes, lack of recurrent Vβ family or CDR3 sequence expansion among patients suggests that clonotypes are private to each disease and that uncommon elusive chronic antigens might drive monoclonal expansion of leukemic cells [74,75]. CDR3 sequences might be shared within patients with marrow failure syndromes and between patients with different diseases and healthy subjects at very low frequencies, as described for paroxysmal nocturnal hemoglobinuria (PNH)-related clonotypes found at very low frequencies in patients with AA and healthy controls [53]. Therefore, TCR clonality in effector memory-like T cells indicates a chronic antigen stimulation in disease initiation; however, whether a common or an elusive infrequent antigen triggers this expansion remains an open question.

## 4. Flow Cytometric Vβ Usage

The diagnostic definition of clonality is still challenging because current technologies alone cannot conclusively define TCR oligo/monoclonality or cannot be applied in routine diagnostic settings [74,75,76]. For example, Southern blot has been the gold standard until a few decades ago, because it can ideally identify every TCR gene rearrangement if appropriate probes and restriction enzymes are used; however, Southern blot is time and labor consuming and high-quality DNA is required [77,78]. PCR cannot cover all possible TCR αβ gene rearrangements and is usually applied for the detection of a limited number of sequences, or for TCR gamma (*TCRG*) rearrangements, which are more limited than those of *TCRB* gene [79,80]. Sanger sequencing and spectratyping of CDR3 are also old-fashioned techniques because of their impracticability to individual sequencing or on a large number of T cell clones [75]. In recent years, deep next-generation sequencing (NGS) has completely changed the analysis of TCR repertoire because a single clone with a unique rearrangement and CDR3 sequence can be detected, even at very low frequency; however, TCR repertoire by NGS is still expensive and requires high-quality DNA and bioinformatics competence for analysis of high-throughput data [53,75,81,82,83]. Therefore, despite its relevance in research settings, TCR repertoire by NGS is not yet applicable to routine diagnostic analysis. The introduction of TCR Vβ monoclonal antibodies has markedly improved diagnostic definition of clonality in LGL leukemia [76,84,85]. Current Vβ antibodies cover more than 65% of all Vβ domains and can be grouped in 22 families (Table 1) [86,87].

In available kits, antibodies are conjugated to fluorescein isothiocyanate (FITC), phycoerythrin (PE), or tandem FITC/PE; therefore, sample acquisition requires instruments equipped just with a blue laser (488 nm excitation) and sensors for detection of FITC and PE emission wavelengths (peaks at 525 nm and 574 nm, respectively) [45,68]. Vβ usage by flow cytometry also allows absolute clone count and simultaneous characterization of TCR Vβ distribution in different lymphocyte subsets using appropriate antibody combinations [74]. For example, TCR Vβ usage has been studied on CD3^+^, CD4^+^, and CD8^+^ T cells [74,88,89] in LGL leukemia patients, or in naïve, effector, and memory lymphocytes in healthy subjects [57], or in total CD4^+^ and CD8^+^, effector CD4^+^CD28^+^ and CD8^+^CD28^+^, and effector memory CD4^+^CD28^−^CD57^+^ and CD8^+^CD28^−^CD57^+^ cells in AA [53,70,71]. When a known immunodominant clone has to be monitored (e.g., during treatment or follow-up), immunophenotype of LGLs can be combined with conjugated antibody corresponding to the expanded Vβ family in a multiparametric staining [74]; however, the fluorophore choice for some Vβ family is constrained, as Vβ4 is conjugated only with tandem FITC/PE, Vβ6.1, Vβ6.7, Vβ12, Vβ13.3, and Vβ20 only with FITC, and Vβ13.2 only conjugated with PE (Figure 1). Antibodies conjugated with fluorophores excited by violet (405 nm) and ultraviolet (320 nm) lasers are available for Vβ3, Vβ5.1, and Vβ8, and conjugated in VioBlue (excitation 400 nm/emission 452 nm) for Vβ2, Vβ11, Vβ13.1, and Vβ21.3. Recombinant antibodies are also available for Vβ2, Vβ5.1, Vβ5.3, Vβ7.1, Vβ7.2, Vβ11, Vβ13.1, Vβ13.6, Vβ14, Vβ16, Vβ17, Vβ21.3, and Vβ23. The advantages of recombinant antibodies are reduced lot-to-lot variability and higher purity, because antibodies are produced using a specific DNA sequence for one type of heavy and light chain using mammalian cell lines. Moreover, constant region is the same specifically mutated IgG1 sequence requiring only one type of isotype control [90].

The presence of 1–3 immunodominant Vβ clones does not prove clonality and the diagnosis of LGL leukemia, because flow cytometry results should be validated by DNA sequencing of both Vα and Vβ regions and supported by clinical manifestations and immunophenotype compatible with LGL leukemia [1,74,91]. Vβ skewing can be also frequently found in elderly (more than 40% of healthy subjects aged 45 years and older) caused by infections occurring during the lifetime without any clinical significance [53,92,93,94]. Therefore, Vβ usage by flow cytometry should be defined based on mean Vβ frequencies detected in a pool of healthy subjects used as a reference range, and skewing should be defined when the frequency of a particular Vβ family is higher than the mean + 3 standard deviations (SDs) in healthy subjects [53,74]. Despite those limitations, flow cytometric Vβ usage analysis could be a rapid and effective screening tool for identification of clonal LGLs, especially for differential diagnosis with other clonal benign disorders or reactive conditions [1,74]. Moreover, once clonality is confirmed by molecular analysis that is time and labor consuming and performed in specialized laboratories, flow cytometry Vβ usage could be employed for monitoring the leukemic clone expansion during treatment and follow-up using a fast and specific technique that is commonly available in the majority of Hematology Centers [74].

## 5. Literature Search

Relevant literature using the key word “large granular lymphocyte leukemia” was searched in PubMed database from 1977 to July 2021. Limiting factors were “large granular lymphocyte leukemia”, and full text available in English language. Two investigators independently reviewed the reference list for potential eligible manuscripts, and selected articles were then reviewed independently for inclusion in the analysis. Studies were included when: (1) the year of publication was between 2011 and 2021; (2) the studies were conducted on humans and reported clinical data and Vβ usage by flow cytometry; and (3) the studies were on T-LGL leukemia (Figure 2).

From selected articles (*n* = 20), data were collected into a standardized form including publication year, source, number of total patients and divided by sex, number of subjects with autoimmune disorders, hematological malignancies, or cancers, and Vβ usage by flow cytometry (Table 2) [41,42,46,75,76,88,89,90,91,92,93,94,95,96,97,98,99,100,101,102,103,104,105,106,107]. A total of 533 T-LGL leukemia patients were evaluable for Vβ usage analysis, and sex was available in 524 subjects: 308 of them were males (58.8%) and 216 females (41.2%) with a male:female ratio of 1.4, as already described in smaller cohorts [1]. Autoimmune disorders were reported in 136 subjects (25.5% of cases) and hematological disorders or cancers in 189 patients (35.5%). RA was diagnosed in 24 subjects (4.5% of total cases and 17.6% of patients with autoimmune disorders), and positivity for anti-nuclear antibodies (ANA) or rheumatoid factor (RF) was reported in 58 (10.9% of total cases and 42.6% of subjects with autoimmune disorders) and 16 subjects (3% of total cases and 11.8% of patients with autoimmune disorders), respectively. Other autoimmune disorders were collagenosis, myositis, autoimmune thyroiditis (*n* = 6; 1% of total cases), and autoimmune polyglandular syndrome type 1. Among hematological diseases, PRCA was the most frequent disorder associated with T-LGL leukemia (*n* = 45; 8.4% of total cases), followed by plasma cell disorders (*n* = 25; 4.7% of total cases), especially monoclonal gammopathy of uncertain significance (MGUS), and myelodysplastic syndromes (MDS; *n* = 21; 3.9% of total cases). Other reported hematological conditions associated with T-LGL leukemia were: AIHA (*n* = 10; 1.9% of total cases); T- and B-cell NHL (*n* = 9; 1.7% of total cases); acute leukemias, both myeloid and lymphoblastic; immune thrombocytopenic purpura (ITP); or AA. Solid neoplasms were also frequently associated with T-LGL leukemia (*n* = 52; 9.8% of total cases). Reported frequencies are slightly higher especially for AIHA, ITP, and RA likely because Vβ usage by flow cytometry is not routinely performed, and TCR clonality is mostly confirmed by PCR without indicating expanded Vβ family but only reporting LGL phenotype and clinical manifestations [18].

## 6. Vβ Usage in T-LGL Leukemia

TCR clonality is a major criterion for LGL leukemia diagnosis and can be defined using a molecular or flow cytometric analysis, as described above, and clonal kinetics have shown clinical utility in monitoring disease progression and responsiveness to treatment [74]. TCR repertoire is heterogeneous among patients, and there is no recurrent Vβ expansion across studies [53,75]. However, because of the lack of a consensus on TCR clonality detection methodology, comparison of Vβ skewing among cohorts, case series, or case reports is difficult, especially when clonality is reported as the nucleotide length distribution by PCR without reporting specific *TCRBV* or *TCRBJ* genes involved. Indeed, among more than 700 studies between 2001 and 2021, only 20 were selected for data comparison of Vβ usage by flow cytometry. When studies were analyzed independently, no recurrent Vβ family was found between LGL leukemia patients. Of total 533 evaluable patients, the presence of at least one immunodominant clone by flow cytometry was described in 265 subjects (49.7%), and 1–3 Vβ subgroups were expanded in 12 of them (4.5% of cases). Among studies, the most frequent expanded Vβ family was the Vβ13.1 (*n* = 33; 12.5% of cases), followed by Vβ3 (*n* = 29; 10.9%), Vβ17 (*n* = 24; 9.1%), and Vβ2 and Vβ14 (both *n* = 22; 8.3% of total cases) (Figure 3). Vβ8 represented the immunodominant clone in 6.8% of cases (*n* = 18), and Vβ1 in 5.7% (*n* = 15). Other Vβ families were expanded in less than 5% of cases, and in less than 1% of subjects (case reports) for Vβ5.2, Vβ5.3, Vβ6.1, Vβ6.7, Vβ11, and Vβ13.3. Of total cases, the immunophenotype of LGLs was detailed only in 520 subjects, and 89 cases (17.1%) were CD4^+^ T-LGLs, 19 (3.7%) CD4^+^CD8^+/dim^, and the remaining cases were CD8^+^ T-LGLs. Of the total CD4^+^ T-LGL leukemia cases, in 48 of them, Vβ usage was described in detail, and there was no Vβ skewing in most cases (*n* = 20; 41.7%) or Vβ13.1 expansion (*n* = 16; 33.3%). Other reported expanded Vβ families were: Vβ3; Vβ8; Vβ17; and anecdotical Vβ1, Vβ2, Vβ5.2/5.3, Vβ12.2, Vβ14, and Vβ22. Of the total 33 cases with expanded Vβ13.1, 16 of them (48.5%) were CD4^+^ T-LGLs, three (9.1%) were CD4^+^CD8^+^, and the remaining 14 cases (42.4%) were CD8^+^ T-LGLs. These cumulative data are similar to those reported in single case series showing a preferential Vβ13.1 expansion in CD4^+^ T-LGL leukemia [46,75]. Vβ13.1 family expansion is recurrent also during infectious diseases, especially CMV and human immunodeficiency virus (HIV) [108,109,110]. In the latter, Vβ13.1 expansion is more frequently found in CD4^+^CD8^dim^ T cells expressing the killer lectin-like receptor NKG2D, a CD4^+^ T cell subset present in RA, human T-lymphotropic virus 1-associated myelopathy, and Wegener granulomatosis [108,111]. Interestingly, these CD4^+^CD8^dim^NKG2D^+^ T cells bearing the HLA-DRB1*0701 allele and restricted to Vβ13.1 have a highly homogeneous TCR repertoire [110]. Moreover, CD4^+^CD8^dim^NKG2D^+^ T cells can produce FasL while protected from Fas-mediated growth arrest because of NKG2D protective functions [112]. Therefore, the presence of Vβ13.1 expansion in CD4^+^ leukemic cells might add evidence to the antigen-driven clonal drift pathogenetic theory in T-LGL leukemia.

An exact association between Vβ usage and underlying hematological disease rather than T-LGL leukemia was identified in only 32 subjects out of 189 patients (16.9% of hematological cases). There was no recurrent Vβ expansion, despite Vβ14 represented the immunodominant clone in six cases (18.8% of hematological cases). Vβ14 was not associated with a specific disease, as it was expanded in subjects with MDS, acute leukemias, NHL, or chronic myeloid leukemia; however, the number of evaluable patients was too small to draw any significant conclusion. Conversely, these data could be a starting point for implementing Vβ usage analysis by flow cytometry in subjects with hematological conditions with concomitant increased LGL count. Indeed, case series report concomitant Vβ skewing in MDS patients [107]; however, the number of studied patients is still limited for identification of a recurrent Vβ expansion in a specific MDS group, or if clonal kinetics might be related to disease progression to acute myeloid leukemia (AML). The concept that Vβ skewing is patient- or disease-specific is probably an evolving concept, especially in the NGS era. Sequencing can identify even one TCR rearrangement, and has shown that, despite clonotypes being private to each disease (e.g., T-LGL leukemia or AA), those expanded CDR3 sequences might be found at very low frequencies even in healthy subjects [53,75]. For example, PNH-related clonotypes have been described at low frequencies in both AA and healthy subjects supporting the hypothesis of an autologous immune attack against a self-antigen on hematopoietic stem cells [53,113].

## 7. Conclusions

LGL leukemia is classified as a lymphoproliferative disorder of mature T or NK cells; however, this clinical entity is considered among other BMF syndromes and is frequently associated with autoimmune disorders and other benign and malignant hematological conditions, such as PRCA or acute leukemias [1,2,3]. The immunophenotype of leukemic cells resembles that typical of effector memory CD8^+^ T cells with aberrancy and TCR clonality, mostly TCR αβ and less commonly TCR γδ clonal rearrangements [1,73]. Demonstration of TCR clonality by molecular and/or flow cytometric analysis together with clinical manifestations and increased LGL count by cytology is required for the correct diagnostic definition of this disease, as LGLs can also be present in reactive conditions without any clinical significance [74,92,93,94,95,96]. Molecular analysis is more accurate and can identify every TCR rearrangement, even at very low frequencies, by using novel deep sequencing technologies (e.g., NGS or TCR repertoire coupled with single-cell RNA sequencing) [75,114]. However, these methods are time and labor consuming and high-quality DNA is required [74,75]. Vβ usage by flow cytometry can identify clonal TCR rearrangements at protein level by recognition of 22 Vβ protein families without differentiating within multiple rearrangements that can occur in each given family [53,75]. Despite flow cytometry not being explored as deeply in TCR repertoire analysis as NGS, this technique is fast, sensitive, and almost always available in every Hematology Center, and can be associated with immunophenotypic characterization of the neoplastic clone in a multiparametric staining [53,70,71,72,74,88,89]. Therefore, Vβ usage by flow cytometry might help in quickly identifying clonality and confirming clinical, cytological, and molecular findings; however, the absence of clonal expansion by flow cytometry cannot exclude TCR clonality that might be detected with more sensitive techniques, such as NGS. Moreover, a lack of Vβ skewing might be related to a small T-LGL clone and Vβ usage analysis performed on total CD3^+^CD8^+^ T cells. Indeed, small T-LGL clonotypes might be diluted on total CD8^+^ T cells that appear polyclonal, as elegantly described in AA by both flow cytometry and NGS [53]. In addition, once identified, the immunodominant clone can be easily monitored during treatment and follow-up and can identify disease progression or relapse early, even under clonal drift, as NGS or TCR sequencing are still expensive techniques that cannot be routinely and frequently applied in clinical practice [74]. Finally, the presence and frequency of LGLs should be routinely checked in patients with autoimmune and hematological diseases because of their frequent association, and Vβ usage by flow cytometry might identify some subgroups of hematological conditions with a profound immune dysregulation that might benefit from immunomodulatory or immunosuppressive therapy [107]. Therefore, Vβ usage analysis should be implemented to better clarify the role of LGLs, either reactive or clonal, in other benign and malignant conditions, and to possibly identify recurrent TCR rearrangements for prediction of possible self-antigens triggering the autologous T cell expansion during LGL leukemia and other autoimmune and hematological or neoplastic diseases.

## Figures and Tables

**Figure 1 cancers-13-04513-f001:**
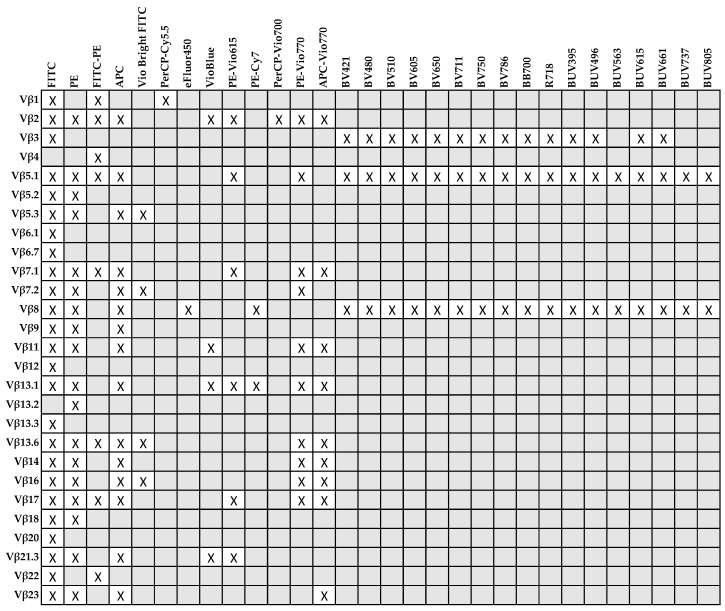
Conjugated antibodies for each Vβ family available for flow cytometry and relative fluorophores (X).

**Figure 2 cancers-13-04513-f002:**
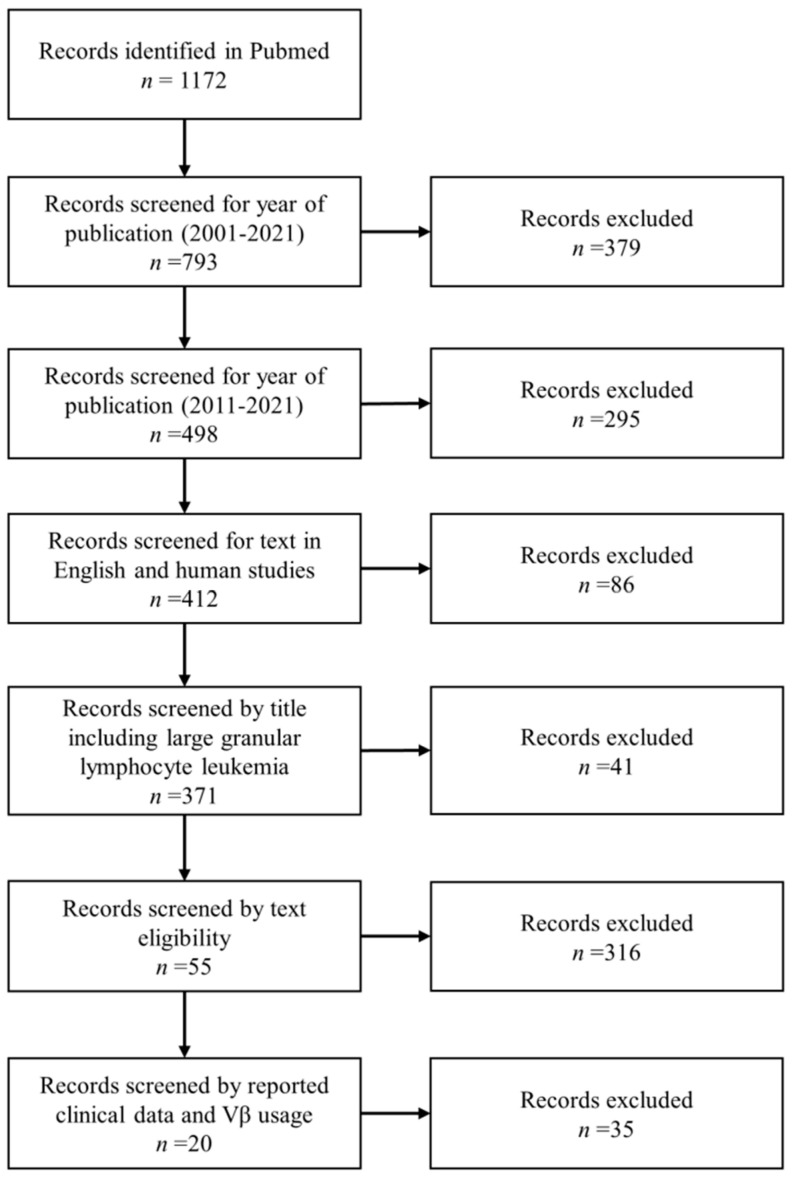
Literature search strategy using “large granular lymphocyte leukemia” as a keyword in Pubmed. After identification of 1172 articles, records were screened by year of publication, title, and text eligibility. A total of 20 articles were eligible for further analysis.

**Figure 3 cancers-13-04513-f003:**
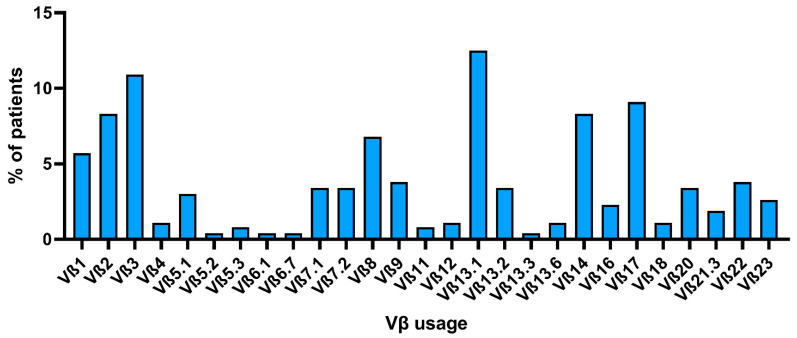
Vβ usage by flow cytometry among selected studies. The percent of patients across selected articles who showed expansion of an LGLL clone carrying each specific Vβ family is reported.

**Table 1 cancers-13-04513-t001:** Antibodies for each Vβ family.

Vβ	Clone	Manufacturer	Vβ (IMGT)
Vβ1	BL37.2	Beckman Coulter (Brea, CA, US), BioLegend (San Diego, CA, US)	TRBV9
Vβ2	MPB2/D5	Beckman Coulter	TRBV20-1
	REA654	Miltenyi Biotec (Auburn, CA, US)	
Vβ3	CH92	Beckman Coulter	TRBV28
	JOVI.3	BD Biosciences (Franklin Lakes, NJ, US)	
	1C1	Invitrogen (Waltham, MA, US)	
	8F10	Abcam (Cambridge, UK)	
Vβ4	WJF24	Beckman Coulter	TRBV29-1
Vβ5.1	LC4	Beckman Coulter	TRBV5-1
	MEM-262	BioLegend	
	IMMU157	BD Biosciences, Invitrogen	
	REA1062	Miltenyi Biotec	
Vβ5.2	4H11	T-cell Sciences (Newburyport, MA, US)	TRBV5-6
	36213	Beckman Coulter	
	MEM-262	BioLegend	
Vβ5.3	4H11	T-cell Sciences	TRBV5-5
	3D11	Beckman Coulter	
	MEM-262	BioLegend	
	REA670	Miltenyi Biotec	
Vβ6.1	CRI304.3	Immunotec (Vaudreuli-Dorion, Canada)	
Vβ6.7	OT145	Gentaur (Kampenhout, Belgium)	
Vβ7.1	Zoe	Beckman Coulter	TRBV4-1, TRBV4-2, TRBV4-3
	ZOE5.1	BioLegend	
	REA871	Miltenyi Biotec	
Vβ7.2	ZIZOU4	Beckman Coulter	TRBV4-3
	REA677	Miltenyi Biotec	
Vβ8	56C5.2	Beckman Coulter	TRBV12-3, TRBV12-4
	JR-2	BioLegend, BD Biosciences. Invitrogen	
	MX-6	Invitrogen	
Vβ9	FIN9	Beckman Coulter	TRBV3-1
	MKB1	BioLegend	
	AMKB1-2	Invitrogen	
Vβ11	C21	Beckman Coulter	TRBV25-1
	REA559	Miltenyi Biotec	
Vβ12	VER2.32	Beckman Coulter	TRBV10-3
	S511	Invitrogen	
Vβ13.1	BAM13	T-cell Sciences	TRBV6-5, TRBV6-6, TRBV6-9
	IMMU222	Beckman Coulter	
	H131	BioLegend, Invitrogen, Abcam	
	REA560	Miltenyi Biotec	
Vβ13.2	H132	Beckman Coulter, BioLegend, Invitrogen	TRBV6-2
Vβ13.3	BAM13	T-cell Sciences	
Vβ13.6	JU74.3	Beckman Coulter	TRBV6-6
	REA554	Miltenyi Biotec	
Vβ14	CAS1.1.3	Beckman Coulter	TRBV27
	REA557	Miltenyi Biotec	
Vβ16	TAMAYA1.2	Beckman Coulter	TRBV14
	REA553	Miltenyi Biotec	
Vβ17	E17.5F3	Beckman Coulter	TRBV19
	REA915	Miltenyi Biotec	
Vβ18	BA62.6	Beckman Coulter	TRBV18
Vβ20	ELL1.4	Beckman Coulter	TRBV30
Vβ21.3	IG125	Beckman Coulter	TRBV11-2
	REA894	Miltenyi Biotec	
Vβ22	IMMU546	Beckman Coulter	TRBV2
Vβ23	AF23	Beckman Coulter	TRBV13
	αHUT7	BioLegend	
	REA497	Miltenyi Biotec	

**Table 2 cancers-13-04513-t002:** Characteristics of included studies.

Study	Year	Journal	Total Cases	M/F	Autoimmune Disorders (*n*)	Hematological Disorders (*n*)
Langerak et al. [76]	2001	Blood	23	13/10	N.R.	N.R.
Stalika et al. [95]	2010	Hematology Oncology	1	1/-		MCL = 1
Garban et al. [96]	2012	Annals of Oncology	12	N.R.		AIL-T = 1
						PTCL-NOS = 1
Hsieh et al. [89]	2012	International Journal of Laboratory Hematology	17	12/5	RF = 2	
					RA = 1	
Koskela et al. [97]	2012	New England Journal of Medicine	77	52/25	RA = 9	PRCA = 5
						MGUS = 5
					Others = 4	CLL = 1
Clemente et al. [75]	2013	Blood	11	7/4	RA = 1	N.R.
Andersson et al. [98]	2013	Blood Cancer Journal	3	2/1	None	MGUS = 1
Rajala et al. [41]	2013	Blood	4	2/2	Collagenosis = 1	AIHA + MGUS = 1
Papalexandri et al. [99]	2013	Bone Marrow Transplantation	2	1/1		AML = 1
						ALL = 1
Stalika et al. [100]	2014	Leukemia and Lymphoma	2	2/-	ANA = 1	NHL = 1
Andersson et al. [42]	2016	Blood	11	4/7		
Singleton et al. [101]	2015	American Journal of Clinical Pathology	54	27/27	RA = 5	NHL = 8
						MDS = 5
					ANA = 1	MPN = 2
						AML = 1
					Others = 2	AA = 1
						Others/cancers = 7
Andersson et al. [88]	2016	Leukemia	4	2/2		Hypergammaglobulinemia = 1
Qiu et al. [102]	2016	Oncotarget	36	20/16	RF = 3	PRCA = 18
					ANA = 7	
Peng et al. [103]	2016	Hematology	10	9/1		PRCA = 10
Awada et al. [104]	2019	British Journal of Haematology	15	9/6		NHL = 3
						ALL = 1
						AA = 1
						CML = 1
Zhu et al. [105]	2020	Leukemia Research	108	51/57	RF = 11	ITP = 3
					RA = 3	PRCA = 22
					ANA = 49	
Minish et al. [106]	2020	Cureus	1	1/-		Melanoma + squamous cell carcinoma = 1
Barilà et al. [46]	2020	Leukemia	129	72/50	Autoimmune thyroiditis = 6	AIHA = 9
						Plasma cell disorders = 11
					RA = 5	MDS = 7
					APS-1 = 2	HCL = 1
					Others = 22	Cancers = 44
Durrani et al. [107]	2020	Leukemia	13	11/2		AML = 4
						MDS = 8
						CMML = 1

Abbreviations. N.R., not reported; MCL, mantle cell lymphoma; AIL-T, angioimmunoblastic T-cell lymphoma; PTCL-NOS, peripheral T-cell lymphoma not-otherwise specified; RF, rheumatoid factor; RA, rheumatoid arthritis; PRCA, pure red cell aplasia; MGUS, monoclonal gammopathy of uncertain significance; CLL, chronic lymphocytic leukemia; AIHA, autoimmune hemolytic anemia; AML, acute myeloid leukemia; ALL, acute lymphoblastic leukemia; ANA, anti-nucleus antibodies; NHL, non-Hodgkin lymphomas; MDS, myelodysplastic syndromes; MPN, myeloproliferative neoplasms; AA, aplastic anemia; CML, chronic myeloid leukemia; ITP, immune thrombocytopenic purpura; HCL, hairy cell leukemia; CMML, chronic myelomonocytic leukemia. APS-1, autoimmune polyglandular syndrome type 1.

## Data Availability

Data are available upon request by the authors.

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
