# Peer review of "The Value of Flow Cytometry Clonality in Large Granular Lymphocyte Leukemia"

_cancers, 2021, doi:10.3390/cancers13184513_

Round 1

Reviewer 1 Report

In this paper, the Authors provide a review of T-Large Granular Cell Leukemia, with a particular focus on biological significance of clonality of LGLs and clinical utility of TCR repertoire investigation by flow cytometry (MFC).

MFC is available in most Hematological facilities. Thus, flow cytometric Vβ usage analysis could be a rapid and effective screening tool for identification of clonal LGLs, especially useful for differential diagnosis with other benign or reactive conditions. Then, after confirmation of clonality through molecular analysis Vβ usage may be useful for a faster and less expensive longitudinal monitoring of the disease.

The review is very well written and clear, and provides useful insights.

Minor:

Please check box 1 and box 2 in Figure 2 (Records identified in Pubmed and Records screened for year of publication 2011-2021), as the first should report the total number of relevant literature retrieved from PubMed database from 1977 to July 2021, whereas the second should report the number of relevant literature in the restricted time frame. The two box currently report the same number (793). The figure caption should be checked as well.

Author Response

Minor

Please check box 1 and box 2 in Figure 2 (Records identified in Pubmed and Records screened for year of publication 2011-2021), as the first should report the total number of relevant literature retrieved from PubMed database from 1977 to July 2021, whereas the second should report the number of relevant literature in the restricted time frame. The two box currently report the same number (793). The figure caption should be checked as well.

Response to Minor Comments. We thank the Reviewer for positive comments and for this correction. We apologize for several typos found in Figure 2, and we have changed numbers accordingly. Figure caption has been corrected as well.

Reviewer 2 Report

In this paper Giudice et al reviewed the value of flow cytometry analysis of clonality in LGL  leukemia. The Authors made a literature research for papers on LGL leukemia which included Vbeta usage flow cytometry, collecting data from 524 patients. They indicate that TCR repertoire is heterogeneous in LGL leukemia patients without a recurrent Vbeta expansion across studies.  From their analysis, the presence of a dominant clone by flow cytometry was reported in 49.7% of cases, which is certainly an  unsatisfying result  to support Vbeta usage for screening, considering that 100% of cases under study were characterized by a clonal population of LGL.  However, I agree with Authors that Vbeta analysis by flow represents indeed a useful parameter for screening in the setting of LGL leukemia. I’d suggest to better address this topic by specifying that clonality can also be indirectly suspected when Vbeta antigens expression was  markedly reduced. This evenience can account for 30% of cases (see Lima et al Am J Pathol. 2001). Data on sensibility and specificity of Vbeta flow as compared to molecular analysis of clonality should be added. To be exhaustive, I’d suggest to discuss the clonal drift phenomenon, as reported by Clemente et al (ref 66) which might limit the usefulness of Vbeta follow up by  flow.

Finally, a  possible limitation of this study is to limit the research to the words “LGL Leukemia” for defining the disease, since proliferations of LGL have been differently named during time, the actual definition being commonly accepted since 2008 WHO classification.

Author Response

In this paper Giudice et al reviewed the value of flow cytometry analysis of clonality in LGL leukemia. The Authors made a literature research for papers on LGL leukemia which included Vbeta usage flow cytometry, collecting data from 524 patients. They indicate that TCR repertoire is heterogeneous in LGL leukemia patients without a recurrent Vbeta expansion across studies.

Comment 1. From their analysis, the presence of a dominant clone by flow cytometry was reported in 49.7% of cases, which is certainly an unsatisfying result to support Vbeta usage for screening, considering that 100% of cases under study were characterized by a clonal population of LGL. However, I agree with Authors that Vbeta analysis by flow represents indeed a useful parameter for screening in the setting of LGL leukemia. I’d suggest to better address this topic by specifying that clonality can also be indirectly suspected when Vbeta antigens expression was markedly reduced. This evenience can account for 30% of cases (see Lima et al Am J Pathol. 2001).

Response to Comment 1. We thank the Reviewer for this comment, and on page 11, lines 363-369, the following text was added “Therefore, Vβ usage by flow cytometry might help in quickly identifying clonality and confirming clinical, cytological, and molecular findings; however, absence of clonal expansion by flow cytometry cannot exclude TCR clonality that might be detected with more sensitive techniques, such as NGS. Moreover, lack of Vβ skewing might be relat-ed to a small T-LGL clone and Vβ usage analysis performed on total CD3+CD8+ T cells. Indeed, small T-LGL clonotypes might be diluted on total CD8+ T cells that appear polyclonal, as elegantly described in AA by both flow cytometry and NGS [53].”

Comment 2. Data on sensibility and specificity of Vbeta flow as compared to molecular analysis of clonality should be added.

Response to Comment 2. We strongly agree with the Reviewer point; however, we do not feel to have exhaustive and comprehensive data to outline sensibility and specificity of Vβ usage by flow cytometry also compared to molecular biology techniques, because this point was out of our review’s purpose, and because we should look back at all available literature and identify potential papers with reported flow and molecular findings consistently detected by either PCR, NGS or other methodologies and matched with related flow data.

Comment 3. To be exhaustive, I’d suggest to discuss the clonal drift phenomenon, as reported by Clemente et al (ref 66) which might limit the usefulness of Vbeta follow up by flow.

Response to Comment 3. On page 11, lines 372-373, the following text was added “even under clonal drift as NGS or TCR sequencing are still expensive techniques that cannot be routinely and frequently applied in clinical practice [74].”

Comment 4. Finally, a possible limitation of this study is to limit the research to the words “LGL Leukemia” for defining the disease, since proliferations of LGL have been differently named during time, the actual definition being commonly accepted since 2008 WHO classification.

Response to Comment 4. We agree with the Reviewer on this observation; for this reason, we restricted our analysis on papers published between 2011 and 2021, after the 2008 WHO classification release.

Reviewer 3 Report

This is a very well written, well edited and gap-filling review on the flow cytometric Vß usage to assess clonality in LGL leukemia and other hematological diseases. However, I have a few comments which suggest improvements to the manuscript.

Major comments:

  1. Table 1: the manufacturer is Miltenyi not Miltenji (corrections warranted at 13 sites)
  2. Figure 2: please check all numbers: if we subtract 17 from 371, we cannot get 412. From this point on, the other numbers in the figure are incorrect.

Minor comments:

Page 1, row 43: µm not µ

Page 4, row 161: routine instead of routinely

Page 9, Figure 3: Why are the columns in different colors? Does it mean anything?

Page 10, row 301: is frequently instead of is a frequently

Page 10, row 316: deep instead of deeper

Author Response

This is a very well written, well edited and gap-filling review on the flow cytometric Vß usage to assess clonality in LGL leukemia and other hematological diseases. However, I have a few comments which suggest improvements to the manuscript.

Major comments

Comment 1. Table 1: the manufacturer is Miltenyi not Miltenji (corrections warranted at 13 sites).

Response to Comment 1. We apologize for this typo and we have changed accordingly.

Comment 2. Figure 2: please check all numbers: if we subtract 17 from 371, we cannot get 412. From this point on, the other numbers in the figure are incorrect.

Response to Comment 2. We apologize for several typos found in Figure 2, and we have changed numbers accordingly. Figure caption has been corrected as well.

Minor comments:

Page 1, row 43: µm not µ. Corrected.

Page 4, row 161: routine instead of routinely. Corrected.

Page 9, Figure 3: Why are the columns in different colors? Does it mean anything? We have changed the graph using one color for all Vβ families for avoiding misleading information.

Page 10, row 301: is frequently instead of is a frequently. Corrected.

Page 10, row 316: deep instead of deeper. Corrected.

Round 2

Reviewer 2 Report

The Authors answered to all inquires